# Screening Method for 22q11 Deletion Syndrome Involving the Use of TaqMan qPCR for *TBX1* in Patients with Conotruncal Congenital Heart Disease

**Felix-Julian Campos-Garcia** [1,*,†] 🄳**, Addy-Manuela Castillo-Espinola** [2,†]**, Carolina-Elizabeth Medina-Escobedo** [1]**,
Juan C. Zenteno** [3,4]**, Julio-Cesar Lara-Riegos** [5]**, Hector Rubio-Zapata** [6]**, David Cruz-Robles** [7] 🄳
**and Ana-Isabel Velazquez-Ibarra** [2]

1    Research Department, Instituto Mexicano del Seguro Social "Ignacio García Tellez",
     Merida City 97155, Mexico
2    Pediatric Cardiology Department, Instituto Mexicano del Seguro Social "Ignacio García Tellez",
     Merida City 97155, Mexico
3    Department of Genetics, Institute of Ophthalmology "Conde de Valenciana", Mexico City 06800, Mexico
4    Department of Biochemistry, Faculty of Medicine, Universidad Nacional Autónoma de México,
     Mexico City 04360, Mexico
5    Faculty of Chemistry, Universidad Autónoma de Yucatán, Merida City 97069, Mexico
6    Faculty of Medicine, Universidad Autónoma de Yucatán, Merida City 97000, Mexico
7    Genomic Laboratory, Molecular Biology Department, National Cardiology Institute Ignacio Chavez,
     Mexico City 14080, Mexico
*    Correspondence: felixcampos@gmail.com
†    These authors contributed equally to this work.

**Abstract:** 22q11.2 deletion syndrome is a phenotypic spectrum that encompasses DiGeorge syndrome (OMIM: 188400) and velocardiofacial syndrome (OMIM: 192430). It is caused by a 1.5–3.0 Mb hem-izygous deletion of locus 22q11.2, which leads to characteristic facies, conotruncal cardiovascular malformations, velopharyngeal insufficiency, T-lymphocyte dysfunction due to thymic aplasia, and parathyroid hypoplasia, and, less frequently, neurological manifestations such as delayed psychomo-tor development or schizophrenia. This study aimed to describe a screening method for the diagnosis of 22q11.2 deletion syndrome in patients with Conotruncal Congenital Heart Disease (CCHD), using qPCR to detect the copy number of the *TBX1* gene in a single DNA sample. A total of 23 patients were included; 21 with a biallelic prediction of the *TBX1* copy number gene and 2 with a monoallelic pre-diction who were suspected to be positive and subjected to MLPA confirmation. One patient (4.34%) with truncus arteriosus CCHD was confirmed to have 22q11.2 deletion syndrome. We propose this approach as a possible newborn screening method for 22q11.2 deletion syndrome in CCHD patients.

**Keywords:** newborn screening congenital heart disease; 22q11.2 syndrome; *TBX1*

## 1. Introduction

22q11.2 deletion syndrome (22q11.2DS) is a phenotypic spectrum that encompasses DiGeorge syndrome (OMIM: 188400) and velocardiofacial syndrome (OMIM: 192430). It is caused by a 1.5–3.0 Mb hemizygous deletion of locus 22q11.2. *TBX1* is a gene encoding a conserved DNA binding domain known as the T-box containing transcription factor. It regulates upstream the differentiation of branchiomeric myogenesis (head and neck muscles), and activates myogenic regulatory factors like Myf5, Mrf4, Myod, Myog [1] and cardiogenic development factors like Isl1, Tbx5, Pax9, Fgf8, Fgf10, Pitx2, Mef2c, and *GATA6* [2]. *TBX1* is expressed in cardiac progenitors of the second heart field, involved in the outflow tract development, right ventricle, and atria. *TBX1* is a dose-dependent transcription factor; over-expression results in congenital heart disease, and developmental delay with or without facial dysmorphism [3]. Haploinsufficiency of *TBX1* is responsible

for the conotruncal cardiovascular malformations in 22q11.2DS. The syndrome has many other physical manifestations, such as characteristic facies, velopharyngeal insufficiency, T-lymphocyte dysfunction due to thymic aplasia, and parathyroid hypoplasia, and, less frequently, neurological manifestations such as delayed psychomotor development or schizophrenia [4].

Congenital heart disease (CHD) is the second most frequent malformation in newborns, after neural tube defects [5,6]. Conotruncal congenital heart disease (CCHD) is an embryonic sub-classification of CHD that originates from defects in the division of the conotruncal outlet, the Tetralogy of Fallot (TOF), truncus arteriosus (TA), the transposition of the great vessels (TGV), double-outlet right ventricle (DORV), interrupted aortic arc type B, and pulmonary atresia with ventricular septal defect (PA) [7,8]. Even though up to 25% of patients with CDH have a CCHD and despite it being a very frequent malformation in 22q11.2DS patients, this genetic condition remains underdiagnosed. *TBX1* has been used for the identification of 22q11.2 deletion syndrome: *TBX1* is located in the typical deletion region (TDR) of the 22q11.2 locus, and is a T-box transcription factor with an expression dependent on genetic doses which results in different congenital defects, especially cardiac outflow development CHD and pharyngeal arch development defects. *TBX1* is commonly deleted in typical and atypical microdeletions, which makes it a good candidate for 22q11.2 microdeletion screening through qPCR [9].

22q11.2DS fluorescent in situ hybridization (FISH) is the gold standard for diagnosing 22q11.2 syndrome; however, it is time-consuming because of the necessity of lymphocyte culture. Multiplex ligation-dependent probe amplification (MLPA) has great specificity (97–99%) and sensitivity (95–99%) in the identification of copy number variants [10]; it also works with ultra-pure DNA samples extracted with the same method and needs normal controls for comparisons in the same run. However, MLPA is time-consuming and costly when used as a mass screening method; therefore, it is usually used as a confirmation method in molecular laboratories. Quantitative polymerase chain reaction (qPCR) is used to quantify copy number variants but requires a confirmatory test such as FISH or MLPA, and this screening method has previously been reported for 22q11.2DS and CCHD [11,12]. However, these approaches require multiple probes. To the best of our knowledge, single *TBX1* TaqMan qPCR has not been used as a screening method for 22q11.2DS in CCHD patients. To diagnose 22q11.2 syndrome in large populations of patients with CCHD, we propose a fast and easy method that uses TaqMan qPCR for *TBX1* and MLPA as a confirmatory test.

## 2. Materials and Methods

Patient diagnosis: We used 75 patients belonging to a cohort with congenital cardiomyopathies at the pediatric cardiology department in the Instituto Mexicano del Seguro Social "Ignacio García Tellez", Merida city, Mexico. All patients were evaluated by a pediatric cardiologist and diagnosis was confirmed through echocardiogram or CT angiography. We selected 24 patients with confirmed CCHD. This study was approved by the local Institutional Review Board (R-2019-3203-019) and investigations were conducted according to the principles expressed in the Declaration of Helsinki [13]. Written informed consent was obtained from all participants.

Sample size: We used the formula for the estimation of proportions in finite populations, with 75 total patients, $p = 0.05$ (CI = 95%), precision (D) = 0.0016, leading to the estimation of a total sample size of 21 participants.

TaqMan technique: This is a type of qPCR based on a three-probe reaction, with two probes that flank the region of interest and the TaqMan probe, which is complementary to an internal segment of the target DNA. This probe is labeled with fluorescent motives, which irradiates light only when is degraded by the Taq polymerase. This fluorescence can be measured through PCR cycling. This technique is highly specific for detecting Copy Number Variants (CNV) and less prone to contamination than Sybr Green qPCR [14].

DNA samples: DNA extraction from whole blood in EDTA to all CCDH patients was performed following standard procedures, using the Pure Link® Genomic DNA extraction kit (Thermofisher®, Waltham, MA, USA, part no. K182002), based on the solid-phase adsorption DNA extraction technique [15]. Through the fluorometric technique, DNA was quantified and its purity was checked. All samples were diluted to a standard concentration of 2 ng/μL. The qPCR test was adapted for CCDH screening [11]: For the determination of copy number variation in the 22q11.2 locus, *TBX1* TaqMan probes were multiplexed in a single reaction with the internal standard *RNaseP*. The reaction contained TaqPath ProAmp Master Mix 1X (Thermofisher®, part no. A30866), 1 μL TaqMan copy number assay *TBX1* (PCR primers $20\times$ and FAM dye-labeled TaqMan MGB probe; Thermofisher® part no. 4400291), 1 μL TaqMan copy number reference assay *RNaseP* (PCR primers and VIC dye-labeled TaqMan TAMRA probe; part no. 4403326), 4 μL of nuclease-free water, and 8 ng of genomic DNA, with a total reaction volume of 20 μL. The samples were loaded into a 48-well plate. Amplifications were performed in a StepOne® thermocycler from Applied Biosystems® using the "Quantification $\Delta C_T$" program mode. The results were exported to the CopyCaller® program (Thermofisher®, version 2.1, Waltham, MA, USA) for $2^{-\Delta\Delta C_T}$ interpretation. DNA from a confirmed 22q11 syndrome patient by Fluorescent in situ hybridization (FISH), was used as a positive control (monoallelic control), and DNA from a healthy patient was used as a negative control (biallelic control). All tests were performed in quadruplicate. The cut-off value for monoallelic (deletion) copy number prediction was set at $2^{-\Delta\Delta C_T}$: < 0.70 [16]. Those patients with results below the cutoff were subjected to multiplex ligation-dependent probe amplification (MLPA).

MLPA: Assays were performed using the SALSA MLPA probe mix P250-B1 DiGeorge kit (MRC-Holland, Amsterdam, The Netherlands) [17]. A total of 40 ng of DNA from each patient and control sample was used, with a purity index greater than 1.8. A positive control for Del22q11.2, a healthy (biallelic), and a blank (no DNA) control were included in each assay. We followed the manufacturer's protocol for DNA denaturalization, hybridization, ligation, and PCR. Capillary electrophoresis was performed using the Applied Biosystems® 3130 Genetic Analyzer and crude data were analyzed using the Coffalyzer software (DB v.140701.0000).

Statistical analyses: The R Studio software was used for $2^{-\Delta\Delta C_T}$ calculation and graphic elaboration (ggplot2 library) [18].

## 3. Results

Patients: A total of 16 male patients (66.66%) and 8 female patients (33.33%) were used. Diagnosis was distributed as follows: Tetralogy of Fallot: 16 patients (66.6%); pulmonary atresia with ventricular septal defect: 4 patients (16.66%); transposition of the great arteries: 3 patients (12.5%); truncus arteriosus: 1 patient (4.1%).

A total of 21 patients with a biallelic copy number prediction ($2^{-\Delta\Delta C_T}$: > 0.70) for *TBX1* were considered negative for 22q11.2DS. One patient was lost because of DNA degradation, and two patients had monoallelic copy number predictions ($2^{-\Delta\Delta C_T}$: < 0.70) for *TBX1*, which were 0.67 and 0.37, respectively (Figure 1).

The DNA of the two patients with a monoallelic prediction was subject to MLPA analysis. The MLPA probes used for the analysis was specific to the 22q11.2 loci. Each sample showed a high peak signal in the DNA denaturation controls (D-fragments) at 88 and 96 nt. One patient showed a reduced peak signal of the 22q11.2 probes: CTCL1-3, HIRA-25, CDC45-1, CLDN5-1, GP1BB-2, TBX1-2, TBX1-7, TXNRD2-9, DGCR8-2, ZNF74-2, KLHL22-2, and MED15-10. Relative copy number values were between 0.51 and 0.56. All probes from control samples and reference probes showed a copy number value of 1 (Figure 2). The other patient showed a normal peak signal with a biallelic conclusion.

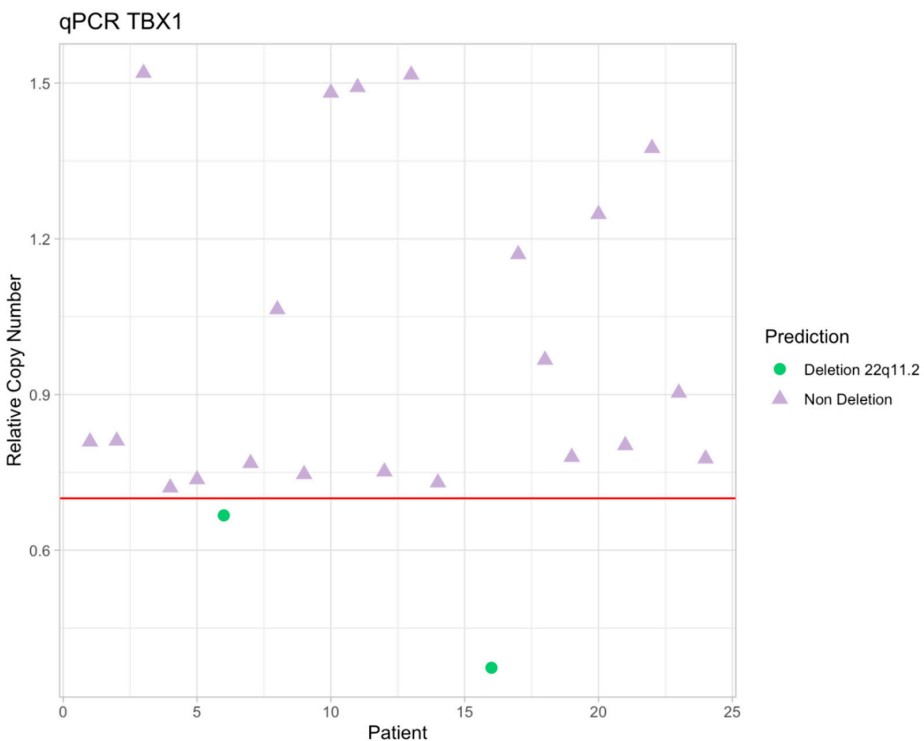

**Figure 1.** Distribution of the relative copy number of TBX1 detected by qPCR. Patients with a $2^{-\Delta\Delta C_T}$: < 0.70 (red line) were considered to possibly have 22q11.2 deletion syndrome (round points) and MLPA analysis candidates.

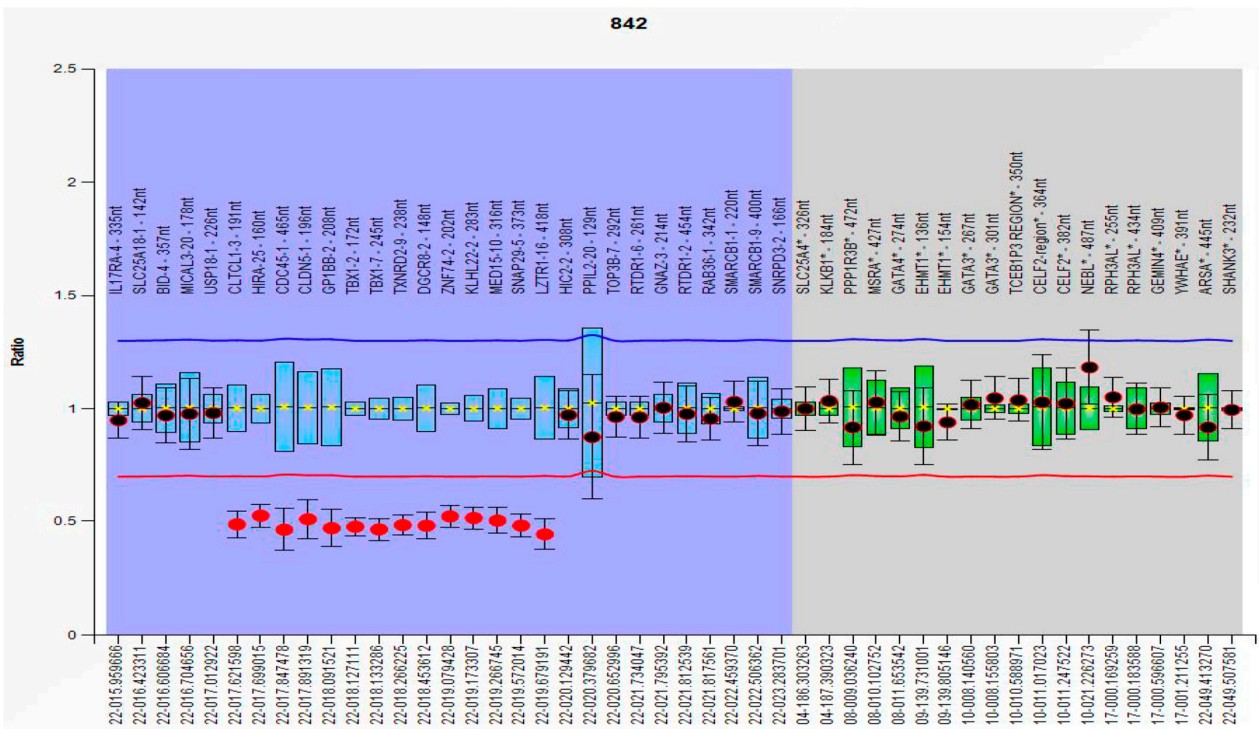

**Figure 2.** Distribution type graph showing the reference samples for the MLPA essay in BioRad2 for a patient positive (ID Lab No. 842) for 22q11.2 deletion syndrome. Red dots under the red line indicate the deletion of CTCL1-3, HIRA-25, CDC45-1, CLDN5-1, GP1BB-2, TBX1-2, TBX1-7, TXNRD2-9, DGCR8-2, ZNF74-2, KLHL22-2, MED15-10, SNAP29, and LZTR1. Black dots between blue and red lines indicate a normal genetic dosage.

Clinical description of a positive 22q11.2DS patient: A 6-month-old girl, a product of a healthy, non-consanguineous couple with a previous abortion with an unknown diagnosis, was referred to the pediatric cardiology department with cardiogenic shock and sepsis. Chest roentgenography showed grade III cardiomegaly and right lung congestion. An electrocardiogram showed biventricular hypertrophy and left atrial enlargement. An echocardiogram showed *situs solitus*, patent *foramen ovale*, *truncus arteriosus*, emergence override by 60%, and ventricular septal defect of 9 × 7 mm. Angiotomography showed four chamber enlargement, 12.4 mm ventricular septal defect, truncus arteriosus type I, and pulmonary stenosis (Figure 3).

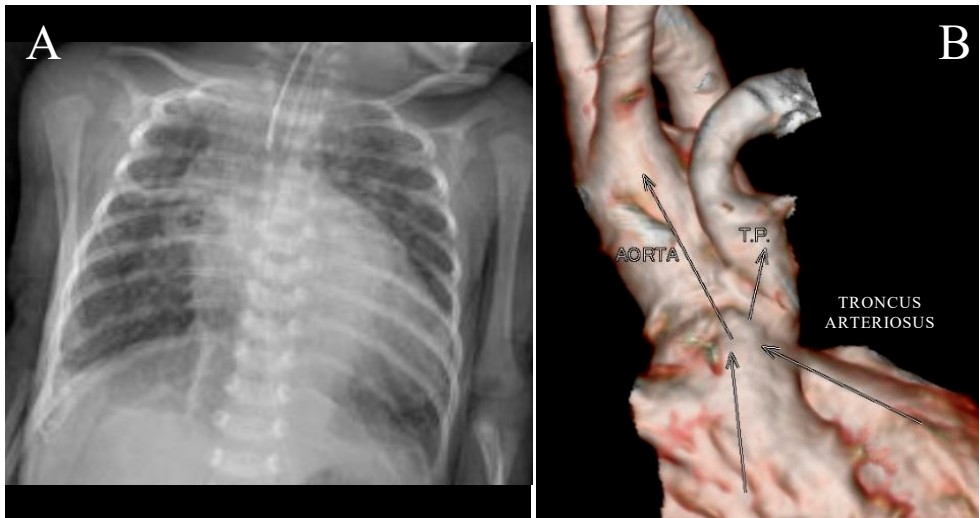

**Figure 3.** Patient positive for 22q11.2 microdeletion syndrome: (**A**) Chest roentgenography showing grade III cardiomegaly and right lung congestion; (**B**) angiotomography showing truncus arteriosus type I and pulmonary stenosis.

## 4. Discussion

CHD is the most common birth defect worldwide [6], with an estimated incidence of 90 per 100,000 newborns affected [5], of which up to 25–30% have CCHD [7]. The etiology of CHD is not clear in most cases; however, 22q11.2DS is described as the second most frequent genetic cause of CHD after Down Syndrome.

*TBX1* encodes a T-box transcription factor, and its haploinsufficiency has been described as the leading cause of the incorrect patterning of the heart outflow tract in 22q11.2 syndrome [19]. However, the exact mechanism by which the minor expression of *TBX1* leads to heart malformations is more complex, involving regulators such as *GATA6* [2] and the Wnt/β-catenin pathway [20]. For this reason, *TBX1* is the ideal target for 22q11.2 screening in patients with CCHD.

We successfully screened the 22q11.2DS in 23 patients using a TaqMan qPCR and MLPA. Even though Tetralogy of Fallot continues to be the most frequent CCHD (66.6%), in this study we observed that the vast majority of patients did not present Del22q11.2. However, a confirmation of one patient positive for 22q11.2DS and the finding of 22q11.2 syndrome occurring a frequency of 4.34% in patients with conotruncal CHD are in agreement with studies that have used similar approaches: Wisconsin, USA (5%) [12], Korea (4.94%) [21], and Malaysia (4.76%) [22] (Table 1).

The diagnosis of 22q11.2DS was not suspected previously in one of our patients, who had a complex CHD with a primary diagnosis of *truncus arteriosus*, patent *foramen ovale*, emergence override, ventricular septal defect, and data showing immunodeficiency as neonatal sepsis. However, its confirmation helped clinicians and geneticists provide adequate treatment and proper genetic counseling for the family. *Truncus arteriosus* is a very complex, poor prognosis CCHD, meaning it is likely that 22q11.2 syndrome is more fre-

quent in complex CCDH, in concordance with other studies that show a high prevalence of 22q11.2 syndrome in complex CCHD [23,24]. However, a bigger and more diversified sample is necessary.

**Table 1.** 22q11.2 deletion syndrome in different screening protocols.

| Population | Screening Method | 22q11.2 Syndrome Frequency | Reference |
|---|---|---|---|
| Congenital Heart Disease | FISH | 14.94% | [25] |
| | qPCR | 5% | [12] |
| | PCR-RFLP, MLPA, FISH | 1.27% | [10] |
| | aCGH | 4.94% | [21] |
| | MLPA | 4.76% | [22] |
| | MLPA | 21.9% | [17] |
| Low T cell receptor excision circles | qPCR, MLPA | 10.7% | [11] |
| Hypocalcemia, cleft lip palate | MLPA | 1.8% | [26] |

Screening for 22q11.2 is complex due to high prices and difficult protocols. Our work advocates for a more simplistic and accessible 22q11.2 screening; a one-probe TaqMan qPCR following a confirmatory study (MLPA), reducing costs and diagnostic times for patients. Screening for 22q11 microdeletion syndrome has been performed previously in children with specific phenotypes, such as multiplexed TaqMan qPCR for CCHD [12], Sybr Green qPCR for CCHD [27], hypocalcemia, cleft palate, and severe combined immunodeficiency [11], using various molecular approaches such as qPCR, qPCR–MLPA, and MLPA. However, these proposed protocols are based on multiplexed qPCR (more than one target probe) or Sybr Green technology, which is less efficient than TaqMan or non-CCHD-specific screening. The implementation of the single-probe TaqMan qPCR 22q11.2DS screening for *TBX1* with ulterior confirmation with MLPA in patients with CCHD provides a useful pathway by which to discern patients with 22q11.2DS from a large population with a single DNA sample and needing no further sampling.

This is a good time for a simple and reliable 22q11.2 syndrome screening that links up patients with treatment and current research, to improve quality of life and prognosis. This study proves that a simplified 22q11.2 screening is feasible and tries to persuade clinicians to boost diagnosis, which is still very low. Evidence shows that 22q11.2 syndrome prognosis is linked to an early diagnosis of CHD or immunodeficiency [28], hence the importance of early 22q11.2 syndrome diagnosis in survival. In the future, we aim for a fast, cheap, and one-step 22q11.2 screening, using amplification-free technologies. In the meantime, we strive for a low-cost array-CGH or Next Generation Sequencing screening; these technologies can give a fast diagnosis of 22q11.2 syndrome, but their cost makes them unsuitable for newborn screening.

Our study had some limitations. The first one was that our patient sample with CCHD was limited, and this was solved by sample size calculation. The second one was that about 3% of 22q11.2 microdeletions are atypical [29], and the single-use of the *TBX1* probe would not have been sufficient to rule out the presence of the deletion in all CCHD patients; thereby, a second probe could be added to qPCR or a different approach with a robust clinical criterion for patient screening could be implemented.

The frequency of the occurrence of 22q11.2DS in southeast Mexico is not different from that in other regions. In our study, we used the same sample for qPCR and MLPA for copy number assay with successful outcomes, as these are less time-consuming than other confirmatory techniques such as FISH. We propose this approach as a possible newborn screening method for 22q11.2 deletion syndrome in CCHD patients.

**Author Contributions:** Methodology, C.-E.M.-E. and J.C.Z.; validation, D.C.-R.; formal analysis, J.-C.L.-R.; investigation, F.-J.C.-G., A.-M.C.-E. and A.-I.V.-I.; writing—original draft preparation, F.-J.C.-G.; supervision, H.R.-Z. All authors have read and agreed to the published version of the manuscript.

**Funding:** This research was funded by CONACYT, grant number 2019-000002-01NACF-13444.

**Institutional Review Board Statement:** The authors assert that all the procedures contributing to this work comply with the ethical standards of the Mexican General Law of Health and with the Helsinki Declaration of 1975, as revised in 2008, and the work has been approved by the "Instituto Mexicano del Seguro Social" local institutional committee of ethics and biosecurity (R-2019-3203-019).

**Informed Consent Statement:** Informed consent was obtained from all the subjects involved in the study.

**Data Availability Statement:** All the data are available upon request.

**Acknowledgments:** We thank Mario Arturo Maldonado Solis (Tamiz Mas Laboratory, Merida, Mexico), collaborator and friend. We will always miss you.

**Conflicts of Interest:** The authors declare no conflict of interest.

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
