# Peer review of "Screening Method for 22q11 Deletion Syndrome Involving the Use of TaqMan qPCR for TBX1 in Patients with Conotruncal Congenital Heart Disease"

_cardiogenetics, doi:10.3390/cardiogenetics12030024_

Round 1

Reviewer 1 Report

The present manuscript is enough clear and presented in a discrete-structured manner.

I appreciate the tecnical pipeline authors use to approach the topic which is enough appropriate.

The cited references are poor for the text of the manuscript. I find necessary to increase the total number of references expecially for the methods in which are missing almost all the references for the used tecniques.

The manuscript could be published with some revisions.

Rows 36-42

Authors should eliminate the templates rules from the text of the manuscript…

Row 68: Authors should put references for MLPA tecnique justifying the use of the tecniques for the kind of approach they decided to use.

Materials and Methods.

Rows 88 to 105: Authors should put references for DNA genomic extraction, Taqman probes, qPCR gene expression analysis and all the other cited tecniques and data analysis tools.

Results

Rows 132-135: Why did authors choose these genes/probes? Authors should  justify the choice and insert references.

Figure 2: Authors should correct the caption of the image/graph: they could eliminate the name (which report the word “ensayo”) of the image inserting all the informations in the caption.

Discussion

Rows 163-165: “However, a confirmation of one patient positive for 22q11.2DS, and a frequency of 4.34% of 22q11.2 syndrome in patients with conotruncal CHD, it´s in agreement with studies  that used similar approaches (Error! Reference source not found.).” Authors must correct the sentence and put the references.

Author Response

Dear Prof: 

We sincerely thank you for your time reviewing our work. Here we list all changes made based on your valuable comments. Attached you will find the response for all reviewers.

REVIEWER 1

The present manuscript is enough clear and presented in a discrete-structured manner.

I appreciate the tecnical pipeline authors use to approach the topic which is enough appropriate.

Reviewer comment

Authors’ response

The cited references are poor for the text of the manuscript. I find necessary to increase the total number of references expecially for the methods in which are missing almost all the references for the used tecniques.

We have added references (also listed in posterior comments):

7. Huber, J.; Peres, V.C.; de Castro, A.L.; dos Santos, T.J.; da Fontoura Beltrão, L.; de Baumont, A.C.; Cossio, S.L.; Dalberto, T.P.; Riegel, M.; Cañedo, A.D.; et al. Molecular Screening for 22Q11.2 Deletion Syndrome in Patients With Congenital Heart Disease. Pediatr. Cardiol. 2014, 35, 1356–1362, doi:10.1007/s00246-014-0936-0.

9. Tomita-Mitchell, A.; Mahnke, D.K.; Larson, J.M.; Ghanta, S.; Feng, Y.; Simpson, P.M.; Broeckel, U.; Duffy, K.; Tweddell, J.S.; Grossman, W.J.; et al. Multiplexed Quantitative Real-Time PCR to Detect 22q11.2 Deletion in Patients with Congenital Heart Disease. Physiol. Genomics 2010, 42A, 52–60, doi:10.1152/physiolgenomics.00073.2010.

11. Aslam, M.M.; John, P.; Fan, K.H.; Bhatti, A.; Feingold, E.; Demirci, F.Y.; Kamboh, M.I. Association of VPREB1 Gene Copy Number Variation and Rheumatoid Arthritis Susceptibility. Dis. Markers 2020, 2020, doi:10.1155/2020/7189626.

12. Boom, R.; Sol, C.J.A.; Salimans, M.M.M.; Jansen, C.L.; Wertheim-Van Dillen, P.M.E.; Van Der Noordaa, J. Rapid and Simple Method for Purification of Nucleic Acids. J. Clin. Microbiol.1990, 28, 495–503, doi:10.1128/jcm.28.3.495-503.1990.

14. Pineda, T.; Zarante, I.; Paredes, A.C.; Rozo, J.P.; Reyes, M.C.; Moreno-Niño, O.M. CNVs in the 22q11.2 Chromosomal Region Should Be an Early Suspect in Infants with Congenital Cardiac Disease. Clin. Med. Insights Cardiol.2021, 15, doi:10.1177/11795468211016870.

16. Adachi, N.; Bilio, M.; Baldini, A.; Kelly, R.G. Cardiopharyngeal Mesoderm Origins of Musculoskeletal and Connective Tissues in the Mammalian Pharynx. Dev. 2020, 147, doi:10.1242/dev.185256.

17. Jiang, X.; Li, T.; Liu, S.; Fu, Q.; Li, F.; Chen, S.; Sun, K.; Xu, R.; Xu, Y. Variants in a Cis-Regulatory Element of TBX1 in Conotruncal Heart Defect Patients Impair GATA6-Mediated Transactivation. Orphanet J. Rare Dis. 2021, 16, 1–14, doi:10.1186/s13023-021-01981-4.

18. Fa, J.; Zhang, X.; Zhang, X.; Qi, M.; Zhang, X.; Fu, Q.; Xu, Z.; Gao, Y.; Wang, B. Long Noncoding RNA Lnc-TSSK2-8 Activates Canonical Wnt/β-Catenin Signaling Through Small Heat Shock Proteins HSPA6 and CRYAB. Front. Cell Dev. Biol. 2021, 9, 1–10, doi:10.3389/fcell.2021.660576.

Rows 36-42

Authors should eliminate the templates rules from the text of the manuscript…

We apologize for this mistake. Thank you.

Row 68: Authors should put references for MLPA tecnique justifying the use of the techniques for the kind of approach they decided to use.

We have added information about MLPA

Multiplex ligation-dependent probe amplification (MLPA) has great specificity (97 – 99%) and sensibility (95 – 99%) to identify copy number variants, [7] also it works with ultra-pure DNA samples extracted with the same method and needs normal controls for comparison in the same run. However, MLPA is time-consuming and costly if it is used as a massive screening method, therefore it is a good confirmation method in a molecular laboratory. Row – 60 - 65

Thank you.

Materials and Methods.

Rows 88 to 105: Authors should put references for DNA genomic extraction, Taqman probes, qPCR gene expression analysis and all the other cited tecniques and data analysis tools.

We have added information about:

DNA genomic extraction: DNA extraction from whole blood in EDTA to all CCDH patients was performed following standard procedures, using the Pure Link® Genomic DNA extraction kit (Thermofisher®, part no. K182002), based on the solid-phase adsorption DNA extraction technique. [12] – Row: 92 - 93

TaqMan probes:

TaqMan Technique: This is a type of qPCR based on a three-probe reaction, two probes that flank the region of interest, and the TaqMan probe that is complementary to an internal segment of the target DNA. This probe is labeled with fluorescent motives, which irradiates light only when is degraded by the Taq polymerase, this fluorescence can be measured through the PCR cycles. This technique is highly specific for detecting Copy Number Variants (CNV) and less prone to contamination than Sybr Green qPCR. [11]

– Row: 84 - 89

qPCR:

qPCR test was adapted for CCDH screening[8]:: - Row: 95

MLPA: Assay was performed by SALSA MLPA probe mix P250-B1 DiGeorge kit (MRC-Holland, Amsterdam, The Netherlands) [14]..Row: 111 - 112

Thank you.

Results

Rows 132-135: Why did authors choose these genes/probes? Authors should  justify the choice and insert references.

All probes used in the MLPA test are specific of the 22q11.2 loci. We have added this sentence to make it clearer:

The MLPA probes used for the analysis are specific for the 22q11.2 loci. – Row: 137

Thank you.

Figure 2: Authors should correct the caption of the image/graph: they could eliminate the name (which report the word “ensayo”) of the image inserting all the informations in the caption.

We apologize for this mistake.

We have eliminated the name in the image and make a full description in the caption.

Figure 2: Distribution type graph showing the reference samples for the MLPA essay in BioRad2 for patient positive (ID Lab No. 842) for 22q11.2 deletion syndrome, red dots under red line indicates deletion of CTCL1-3, HIRA-25, CDC45-1, CLDN5-1, GP1BB-2, TBX1-2, TBX1-7, and TXNRD2-9, DGCR8-2, ZNF74-2, KLHL22-2, MED15-10, SNAP29, LZTR1.” Row- 145 - 148

Thank you.

Discussion

Rows 163-165: “However, a confirmation of one patient positive for 22q11.2DS, and a frequency of 4.34% of 22q11.2 syndrome in patients with conotruncal CHD, it´s in agreement with studies  that used similar approaches (Error! Reference source not found.).” Authors must correct the sentence and put the references.

We apologize for this mistake.

The broken link was for the Table 1. We have corrected the text and the caption of table 1. – Row 179

Table 1. 22q11.2 deletion syndrome in different screening protocols. – Row: 183

Thank you.

Reviewer 2 Report

The authors examine the patients for 22q11.2 deletion syndrome conotruncal congenital heart disease using qPCR and identify the TBX1 in a single patient DNA sample. This study provides an important diagnostic tool to screen the TBX1-associated congenital complications and helps the field. 

Concerns:

1. Need to add more about the TBX1, and its regulators and binding partners in this particular syndrome in the discussion part.

2. This study has methodological limitations, please add it in the discussion section

Author Response

Dear Prof:

We sincerely thank you for your time reviewing our work. Here we list all changes made based on your valuable comments. Attached you will find the response for all reviewers.

REVIEWER 2

The authors examine the patients for 22q11.2 deletion syndrome conotruncal congenital heart disease using qPCR and identify the TBX1 in a single patient DNA sample. This study provides an important diagnostic tool to screen the TBX1-associated congenital complications and helps the field. 

Reviewer comment

Authors’ response

Concerns:

1. Need to add more about the TBX1, and its regulators and binding partners in this particular syndrome in the discussion part.

We have added a paragraph (with 3 citations) about this important information,

TBX1 encodes a T-box transcription factor, its haploinsufficiency has been described as the leading cause of incorrect patterning of the heart outflow tract in 22q11.2 Syndrome [16]. However, the exact mechanism that minor expression of TBX1 leads to heart malformations is more complex, involving regulators like GATA6 [17] and Wnt/β-catenin pathway [18]. For this reason, TBX1 is the ideal target for 22q11.2 screening in patients with CCHD.

– Row: 168 - 173

Thank you.

2. This study has methodological limitations, please add it in the discussion section

To improve our method description, we have added the sample size calculation method to the manuscript. This has not been considered previously, because it is a descriptive study. However, thanks to your observation, we have decided to fully describe it on the manuscript.

Sample Size: We used the formula for the estimation of proportions in finite populations, with 75 total patients, p=0.05 (CI=95%), precision (D)=0.0016. Estimating a total sample size of 21 participants. – Row: 81 - 83

We have also added information on de discussion section:

Our study has two limitations, the first one is that our patient universe with CCHD is limited, this was solved by sample size calculation. The second one is that about 3% of 22q11.2 microdeletions are atypical[24], and the single use of the TBX1 probe would not be enough to rule out the deletion in all CCHD patients, thereby a second probe could be added to qPCR or implement a different approach with a robust clinical criterion for patient screening. – Row: 195 - 200

Thank you.

Reviewer 3 Report

cardiogenetics-1737690

In this work, the author discussed a qPCR screening method for 22q11 DS. Despite the clinical relevance and potential of the work, the reviewer was not convinced by the novelty and quality of the manuscript. Although the results suggest that the Taqman assay could be used for disease screening, similar results have been published before (https://doi.org/10.1152/physiolgenomics.00073.2010). The article was also not well written as numerous obvious errors in grammar and format are noted. 

Author Response

Dear Prof:

We sincerely thank you for your time reviewing our work. Here we list all changes made based on your valuable comments. Attached you will find the response for all reviewers.

REVIEWER 3

In this work, the author discussed a qPCR screening method for 22q11 DS. Despite the clinical relevance and potential of the work, the reviewer was not convinced by the novelty and quality of the manuscript.

Reviewer comment

Authors’ response

Although the results suggest that the Taqman assay could be used for disease screening, similar results have been published before (https://doi.org/10.1152/physiolgenomics.00073.2010).

We have added this valuable citation on the manuscript:

Screening for 22q11 microdeletion syndrome has been performed previously in children with specific phenotypes, such as multiplexed TaqMan qPCR for CCHD [9] Row: 185 - 186

We also contrast the valuable contribution of Prof. Tomita with this work:

Discussion:

However, those proposed protocols were based on multiplexed qPCR (more than one target probe), or based on Sybr Green technology, which is less efficient than TaqMan or not CCHD-specificscreening. Implementation of single-probe TaqMan qPCR 22q11.2DS screening for TBX1 with ulterior confirmation with MLPA in patients with CCHD provides a useful pathway to recognize patients with 22q11.2DS from a large population and with a single DNA sample.– Row: 188 - 194

Thank you.

  The article was also not well written as numerous obvious errors in grammar and format are noted. 

We have made major grammar revision, changed many errors, and made style corrections.

Thank you.

Round 2

Reviewer 1 Report

In this version, authors have incorporated all the suggestions by reviewers.

Introduction: contents are adequate but I think the form could be improved finding a more organic way to link all the parts

Materials and Methods: could it be possible to introduce the single parts with subtitles?

Discussione: as I purposed above, could it be possible to link all the parts in a more organic and discorsive form, also with a brief look at future perspectives or next steps to improve the approach?

Author Response

In this version, authors have incorporated all the suggestions by reviewers.

Reviewer comment

Authors’ response

Introduction: contents are adequate but I think the form could be improved finding a more organic way to link all the parts.

Materials and Methods: could it be possible to introduce the single parts with subtitles?

We have used the MDPI English edition service. In this new draft, major style corrections have been made, including grammar and fluidness.

We have followed the MDPI standard format and sent the draft for style corrections. From our point of view, it is out of format style to make this correction.

Thank you.

Discussione: as I purposed above, could it be possible to link all the parts in a more organic and discorsive form, also with a brief look at future perspectives or next steps to improve the approach?

We have added a new paragraph for future directions.

“This study proves that a simplified 22q11.2 screening is feasible and tries to persuade clinicians to boost diagnosis, which is still very low. In the future, we aim for a fast, cheap, and one-step 22q11.2 screening, using amplification-free technologies. In the meantime, we strive for a low-cost array-CGH or Next Generation Sequencing screening, these technologies can give a fast diagnosis of 22q11.2 syndrome, but their cost makes them unsuitable for newborn screening. “

Thank you.

Reviewer 3 Report

The change made by the authors did not significantly improve the quality of the manuscript. The key problem is the lack of novelty and significance, as a very similar study (https://doi.org/10.1152/physiolgenomics.00073.2010) has been performed with a much larger sample size 12 years ago. The reviewer can not approve the submission. 

Author Response

We respectfully submit that our work is a distinct and novel contribution.

Of course, the work of Prof. Tomita is very valuable and marked the beginning of the 22q11.2 screening. However, it is a very complex protocol and not “real-life adapted” to reach our goal to screen all 22q11.1 patients.

What is important is whether this contribution really advances our knowledge and confidence in this area, and we think it does. We test a simpler, fast, and cheaper protocol that gives good results, especially in low-income areas, like ours. And therefore, it is novel.